# Retention of deposited ammonium and nitrate and its impact on the global forest carbon sink

Geshere Abdisa Gurmesa [1,25], Ang Wang[1,2,3,25], Shanlong Li[1,4], Shushi Peng [5✉], Wim de Vries [6], Per Gundersen [7], Philippe Ciais[8,9], Oliver L. Phillips [10], Erik A. Hobbie[11], Weixing Zhu [12], Knute Nadelhoffer [13], Yi Xi [5], Edith Bai [14], Tao Sun[1], Dexiang Chen[15], Wenjun Zhou[16], Yiping Zhang[16], Yingrong Guo[17], Jiaojun Zhu [1,2], Lei Duan [18], Dejun Li[19], Keisuke Koba [20], Enzai Du [21], Guoyi Zhou[22], Xingguo Han[23], Shijie Han[24] & Yunting Fang [1,2,3✉]

The impacts of enhanced nitrogen (N) deposition on the global forest carbon (C) sink and other ecosystem services may depend on whether N is deposited in reduced (mainly as ammonium) or oxidized forms (mainly as nitrate) and the subsequent fate of each. However, the fates of the two key reactive N forms and their contributions to forest C sinks are unclear. Here, we analyze results from 13 ecosystem-scale paired $^{15}$N-labelling experiments in temperate, subtropical, and tropical forests. Results show that total ecosystem N retention is similar for ammonium and nitrate, but plants take up more labelled nitrate ($20^{25}_{15}$%) ($\mathrm{mean}^{\mathrm{maximum}}_{\mathrm{minimum}}$) than ammonium ($12^{16}_{8}$%) while soils retain more ammonium ($57^{65}_{49}$%) than nitrate ($46^{59}_{32}$%). We estimate that the N deposition-induced C sink in forests in the 2010s is $0.72^{0.96}_{0.49}$ Pg C yr$^{-1}$, higher than previous estimates because of a larger role for oxidized N and greater rates of global N deposition.

[1] CAS Key Laboratory of Forest Ecology and Management, Institute of Applied Ecology, Chinese Academy of Sciences, Shenyang, China. [2] Qingyuan Forest CERN, Chinese Academy of Science, Shenyang, China. [3] Key Laboratory of Isotope Techniques and Applications, Shenyang, China. [4] Institute of Agricultural Resource and Environment, Jilin Academy of Agricultural Science, Changchun, China. [5] Sino-French Institute for Earth System Science, College of Urban and Environmental Sciences, Peking University, Beijing, China. [6] Wageningen University and Research, Environmental Systems Analysis Group, Wageningen, the Netherlands. [7] Department of Geosciences and Natural Resource Management, University of Copenhagen, Copenhagen, Denmark. [8] LSCE (CEA CNRS UVSQ UPSaclay) Centre d'Etudes Orme des Merisiers, Gif sur Yvette, France. [9] Climate and Atmosphere Research Center (CARE-C), The Cyprus Institute, Nicosia, Cyprus. [10] School of Geography, University of Leeds, Leeds, UK. [11] Earth Systems Research Center, Morse Hall, University of New Hampshire, Durham, NH, USA. [12] Department of Biological Sciences, Binghamton University, The State University of New York, Binghamton, NY, USA. [13] Department of Ecology and Evolutionary Biology, University of Michigan, Ann Arbor, MI, USA. [14] School of Geographical Sciences, Northeast Normal University, Changchun, China. [15] Institute of Tropical Forestry, Chinese Academy of Forestry, Guangzhou, China. [16] CAS Key Laboratory of Tropical Forest Ecology, Xishuangbanna Tropical Botanical Garden, Chinese Academy of Sciences, Mengla, China. [17] Jiangxi Provincial Bureau of Forestry, Nanchang, Jiangxi, China. [18] State Key Laboratory of Environmental Simulation and Pollution Control, School of Environment, Tsinghua University, Beijing, China. [19] Institute of Subtropical Agriculture, Chinese Academy of Sciences, Changsha, China. [20] Center for Ecological Research, Kyoto University, Shiga, Japan. [21] State Key Laboratory of Earth Surface Processes and Resource Ecology, and Faculty of Geographical Science, Beijing Normal University, Beijing, China. [22] Institute of Ecology and School of Applied Meteorology, Nanjing University of Information Science and Technology, Nanjing, China. [23] State Key Laboratory of Vegetation and Environmental Change, Institute of Botany, Chinese Academy of Sciences, Beijing, China. [24] School of Life Sciences, Henan University, Kaifeng, China. [25] These authors contributed equally: Geshere Abdisa Gurmesa, Ang Wang. ✉email: speng@pku.edu.cn; fangyt@iae.ac.cn

Human activities have greatly accelerated reactive N emissions to the atmosphere and have increased rates of N deposition globally[1–3]. Depositional fluxes of the two dominant forms of deposited N ($NH_x$ and $NO_y$) are unevenly distributed in space and changing over time[4,5]. Increased N deposition enhances ecosystem carbon (C) sinks[6,7], decreases biodiversity[8], and increases N leaching losses, leading to downstream eutrophication and acidification[9]. Critically, the impacts of N deposition depends on the fate of N inputs to ecosystems, i.e., its retention in biomass and soil organic matter pools versus its losses via leaching and denitrification. The fates of deposited N are expected to differ depending on whether it is in reduced (mainly as ammonium) or oxidized (mainly as nitrate) form. Trees, especially conifers, have often been shown to take up more ammonium than nitrate[10–12], probably because the energetic cost is higher for nitrate assimilation[11,13] and ammonium is more abundant in forest soil[14].

However, field-scale $^{15}N$-tracer experiments are needed to unambiguously quantify and understand how deposited ammonium and nitrate are retained in plant and soil pools. Because of few ecosystem-scale $^{15}N$-labelling experiments in forests[15], the differential fates of deposited ammonium and nitrate in forests from the tropics to high-latitude are currently unknown and thus the latitudinal pattern of retention of ammonium and nitrate and its consequences for forest C sinks remain uncertain. Given the large spatial variation (Fig. 1a) in reduced and oxidized N

deposition across forest biomes[16–18], elucidating the fates of ammonium and nitrate is essential for reliably scaling up the contribution of N deposition to the global C sink.

In this study, we analyze results from ecosystem-scale paired $^{15}N$-tracer (separate additions of $^{15}N$-labelled ammonium vs nitrate as $^{15}NH_4^+$ and $^{15}NO_3^-$, respectively) experiments at 13 forests across tropical, subtropical, warm temperate, and cool temperate climate regions to investigate the fate of atmospheric N deposited in ammonium and nitrate form into the forests. We found a similar total N retention fraction for the two forms, but different allocation to plant and soil pools. Nitrogen retention in plants and soils across sites is predicted by a combination of climate, plant, and soil variables. Using a stoichiometric upscaling approach[19], we estimate a greater N deposition-induced C sink in forests than previous estimates and with greater C gain per unit N deposited in oxidized than in reduced forms.

## Results and discussion

**Differential fates of added $^{15}NH_4^+$ and $^{15}NO_3^-$ in forest ecosystems.** At each site, recovery of $^{15}NO_3^-$ than $^{15}NH_4^+$ was measured for plant and soil pools 1 year after $^{15}N$-labelling (Supplementary Fig. 1 for the pathways of $^{15}N$ partitioning into different ecosystem pools). The results indicate that recovered $^{15}NO_3^-$ and $^{15}NH_4^+$ were distributed differently among ecosystem pools, with significantly higher (paired $t$-test, $n = 13$, $p < 0.001$) retention in plants for $^{15}NO_3^-$ ($20_{15}^{25}\%$) (mean$_{minimum}^{maximum}$,

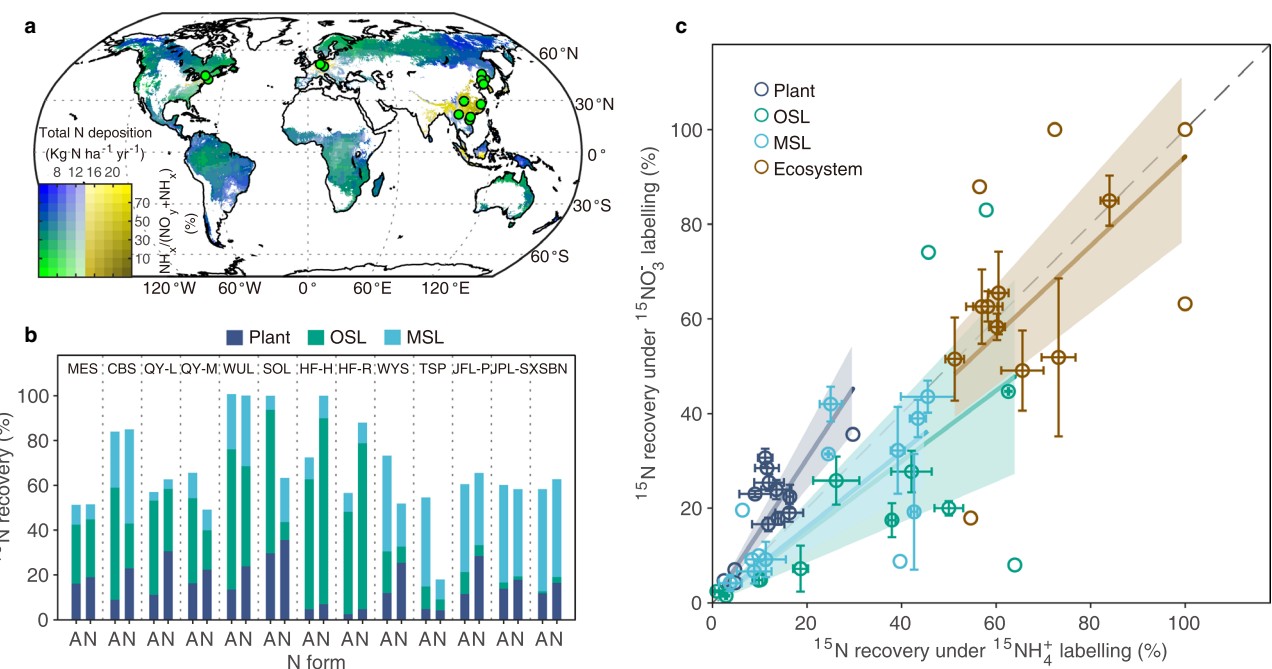

**Fig. 1 $^{15}N$ recoveries under paired $^{15}NH_4^+$ and $^{15}NO_3^-$ labelling across the 13 sites ~1 year after $^{15}N$-labelling. a** Global distribution of total N deposition in forests, the ratio of $NH_x$ to total N deposition in 2010, and locations of the 13 paired $^{15}N$-labelling sites (green circular dots). Global $NH_x$ and $NO_y$ deposition were estimated based on results of four different models (see method) and the average spatial pattern of N deposition is shown. The mapped forested area (42 million km[2]) is derived from global forest cover data[34]; **b** Percent recoveries in plants, organic soil layer (OSL), and mineral soil (MSL) for $^{15}NH_4^+$-labelling (A, ammonium) and $^{15}NO_3^-$-labelling (N, nitrate) at the 13 sites. Abbreviated site names are shown above the bars for Maoershan (MES), Changbai forest (CBS), Qingyuan-larch forest (QY-L), Qingyuan-mixed forest (QY-M), Wülfersreuth (WUL), Solling (SOL), Harvard Forest-hardwood forest (HF-H), Harvard Forest-red pine forest (HF-R), Wuyishan (WYS), Tieshanping (TSP), Jianfengling-primary forest (JFL-P), Jianfengling-secondary forest (JFL-S), and Xishuangbanna (XSBN). The sites are ordered from lowest (left) to highest (right) latitude; **c** Relationship between mean $^{15}N$ recovery for $^{15}NH_4^+$ tracer ($x$-axis) and $^{15}NO_3^-$ tracer ($y$-axis). The black dash line represents the 1:1 line between $^{15}N$ recovery for $^{15}NH_4^+$ tracer and $^{15}NO_3^-$ tracer. Error bars indicate SE for the Chinese sites ($n = 4$ CBS and $n = 3$ for the rest) and WUL ($n = 5$). Error bars are not shown for TSP, HF-H, HF-R, and SOL because only mean values are reported for these sites. In both **b** and **c**, $^{15}N$ recovery shown for plants include both trees and understory (shrubs, herbs, and grasses). In **c**, mean $^{15}N$ recovery in each pool at each site is shown with the shaded areas with the corresponding colour indicating the 95% confidence interval of linear regression for each pool. Slopes of the relationship for plants, organic soil layer, mineral soil layer, and the whole ecosystem are 1.56, 0.79, 0.79, and 1.02, respectively.

95% confidence interval) than for $^{15}NH_4{}^+$ ($12_8^{16}\%$) across sites and plant growth forms (Fig. 1c, Supplementary Fig. 2). Recovery in the soil organic layer was $25_{10}^{39}\%$ for $^{15}NO_3{}^-$ and $33_{21}^{45}\%$ for $^{15}NH_4{}^+$ while recovery in mineral soil was $21_{13}^{29}\%$ for $^{15}NO_3{}^-$ and $24_{15}^{33}\%$ for $^{15}NH_4{}^+$, with a total soil retention of $46_{32}^{59}\%$ for $^{15}NO_3{}^-$ and $57_{49}^{65}\%$ for $^{15}NH_4{}^+$ (Fig. 1b), indicating that both deposited ammonium and nitrate are substantially retained in soil pools. Differences in retention between the two N forms were not statistically significant in the soil organic layer (paired $t$-test, $n = 13$, $p = 0.20$), mineral soil (paired $t$-test, $n = 13$, $p = 0.74$) and for the total soil pool (paired $t$-test, $n = 13$, $p = 0.10$) (Supplementary Fig. 3). The total ecosystem recovery was similar between the two N forms, $67_{55}^{80}\%$ for $^{15}NO_3{}^-$ and $69_{60}^{78}\%$ for $^{15}NH_4{}^+$ (paired $t$-test, $n = 13$, $p = 0.75$) (Fig. 1b, Supplementary Fig. 3), which contrasts with the common assumption that ecosystems retain ammonium more strongly than nitrate[20,21].

The greater recovery of $^{15}NO_3{}^-$ than $^{15}NH_4{}^+$ in plants suggests that deposited nitrate is taken up by plants proportionally more than ammonium. This is not intuitive since plants are viewed to favour ammonium[12,22] because it is generally more abundant in forest soil (Supplementary Table 2) and because both uptake (against a steep electrochemical gradient) and assimilation of nitrate (reduction to $NH_4{}^+$) are energetically more expensive than that of ammonium[11,13] unless nitrate is reduced in leaves under light-saturated conditions[23]. Less plant uptake of deposited ammonium in our experiment could be attributed to preferential assimilation and retention of ammonium by microbes and abiotic mechanisms such as soil adsorption[20,24], which results in ammonium being retained on soil particles. In contrast, nitrate is mobile in soil solution and hence it moves more easily than ammonium to the root surface by diffusion and mass flow[14,25]. This makes nitrate more readily available for plant uptake and could be the main reason for the greater uptake of deposited nitrate by plants in our experiments. While the added $^{15}NO_3{}^-$ was less diluted in the soil than $^{15}NH_4{}^+$, due to the nitrate pool being smaller than the ammonium pool in 11 out of 13 sites (Supplementary Table 2), differences across the study sites in the background soil ammonium to nitrate ratio were unrelated to plant biomass $^{15}N$ recovery ratio for $^{15}NO_3{}^-$ and $^{15}NH_4{}^+$ (Supplementary Fig. 4). This indicates that the consistently higher uptake of deposited nitrate than ammonium by plants across a range of soil ammonium to nitrate conditions was not caused by the greater dilution of exogenous $^{15}NH_4{}^+$ by soil endogenous ammonium.

**Variation of N retention across sites.** Our $^{15}N$ recovery data show that the partitioning to different ecosystem compartments and the total ecosystem retention of deposited ammonium and nitrate both vary across forests (Fig. 1b, Supplementary Fig. 3). In general, plants retained similar amounts of N in either form in both temperate and tropical forests. However, the two N forms experienced different biome-specific fates in soils (Supplementary Fig. 5). The soil organic layer retained significantly more ammonium and nitrate in temperate forests while the mineral soil retained more ammonium and nitrate in tropical forests. The higher N retention in organic soil in temperate forests is likely due to the lower precipitation and thicker organic soil layer in temperate regions than in the tropics[26,27] that provide longer contact time for new N input to be immobilized in the soil organic layer. In tropical forests where the organic layer is usually poorly developed, new N input is likely transported to the mineral soil. In addition, a greater anion adsorption capacity of highly weathered tropical mineral soils could increase retention of the highly leachable nitrate in tropical mineral soil[28]. Ecosystem

retention (soil plus plant) was also greater in temperate forests than in tropical forests for nitrate ($t$-test, $p < 0.05$) (Supplementary Fig. 5), possibly because the higher N availability and precipitation in tropical forests led to higher leaching losses of deposited nitrate. These insights from our paired $^{15}N$-tracer results suggest that varied assimilation and retention efficiency of inorganic N inputs among forest biomes are mainly due to differences in climate, N status, and soil attributes.

We explored the relationship between $^{15}N$ recovery dynamics and sets of factors that potentially predict the capacity of forest ecosystems to retain N (Supplementary Table 4, see Method). We assessed N status from the C/N ratio of the mineral soil and hypothesized that N-limited ecosystems with higher C/N ratios would retain more deposited N than N-rich sites with low C/N ratios[9]. In support of this hypothesis, the fraction of $^{15}NH_4{}^+$ and $^{15}NO_3{}^-$ lost was negatively correlated with the soil C/N ratio ($R^2 = 0.43$, $p < 0.05$ for $^{15}NH_4{}^+$ and $R^2 = 0.32$, $p < 0.05$ for $^{15}NO_3{}^-$) (Fig. 2a). For $^{15}NO_3{}^-$, the fraction of $^{15}N$ lost was also positively correlated with N deposition (Supplementary Fig. 6a). The fraction of $^{15}N$ retained in the organic soil layer was positively correlated with its mass ($R^2 = 0.73$, $p < 0.001$ for $^{15}NH_4{}^+$ and $R^2 = 0.51$, $p < 0.01$ for $^{15}NO_3{}^-$) (Fig. 2b), suggesting that plant-derived soil organic matter strongly controls N retention[29,30]. For $^{15}NH_4{}^+$, retention in organic soil was also negatively correlated with mean annual temperature (MAT) (Supplementary Fig. 6b) and the ratio of $^{15}N$ recovery in plants to that in plants plus mineral soil was negatively correlated with net primary production (NPP) ($R^2 = 0.64$, $p < 0.01$ for $^{15}NH_4{}^+$ and $R^2 = 0.63$, $p < 0.01$ for $^{15}NO_3{}^-$) (Fig. 2c). This negative correlation between NPP and $^{15}N$ recovery in plants relative to the total recovery in plant and mineral soil implies that plant growth (and NPP) in N-rich productive ecosystems including most tropical forests is less dependent on external N from deposition than in N-limited temperate and boreal forests[31,32]. With NPP primarily controlled by climate[33], the correlation could be an indirect climatic control, but MAT or MAP did not appear as important predictors of plant- and mineral soil-related N retention (Supplementary Table 4). In addition, the fraction of $^{15}N$ retained in plants that was allocated to woody biomass varied from 20 to 60% and increased with woody biomass ($R^2 = 0.86$, $p < 0.001$ for $^{15}NH_4{}^+$ and $R^2 = 0.87$, $p < 0.001$ for $^{15}NO_3{}^-$) (Fig. 2d).

**Global deposition and retention pattern of deposited $NH_x$ and $NO_y$ in forest ecosystems.** Despite extensive monitoring and modelling efforts to assess global patterns of N deposition, attempts to separate N deposition rates and its forms for the world's forests have been limited. To quantify global inorganic N deposition to forests and its speciation between reduced and oxidized forms, we used results from four different models (Supplementary Table 5). Total inorganic N deposition on Earth's 42 million $km^2$ of forest[34] in 2000 to 2010 averaged 24.9 Tg N $yr^{-1}$, of which 13.8 Tg N $yr^{-1}$ was $NH_x$ and 11.2 Tg N $yr^{-1}$ was $NO_y$ (Supplementary Table 6). N deposition was higher in tropical (13.1 Tg N $yr^{-1}$) than temperate (8.1 Tg N $yr^{-1}$) and boreal forests (3.7 Tg N $yr^{-1}$) (Supplementary Table 6), although high N deposition occurs in many temperate forests in Asia and central Europe (Fig. 1a). High N deposition was associated with a greater $NH_x$ contribution (Fig. 1a). We used relationships linking plants and soils and the variables described above (NPP, woody biomass, organic layer mass, and soil C/N ratio) to extrapolate the retention of deposited N in global forests (Fig. 2). We then created global maps for the fractions of deposited $NH_x$ and $NO_y$ retained in plants and soil or lost from the ecosystem (Fig. 3, see Methods). We used average values for modelled N deposition (Fig. 1a) and the extrapolated N retention fraction in plant and soil pools (Fig. 2) for both $NH_x$ and $NO_y$ to estimate that

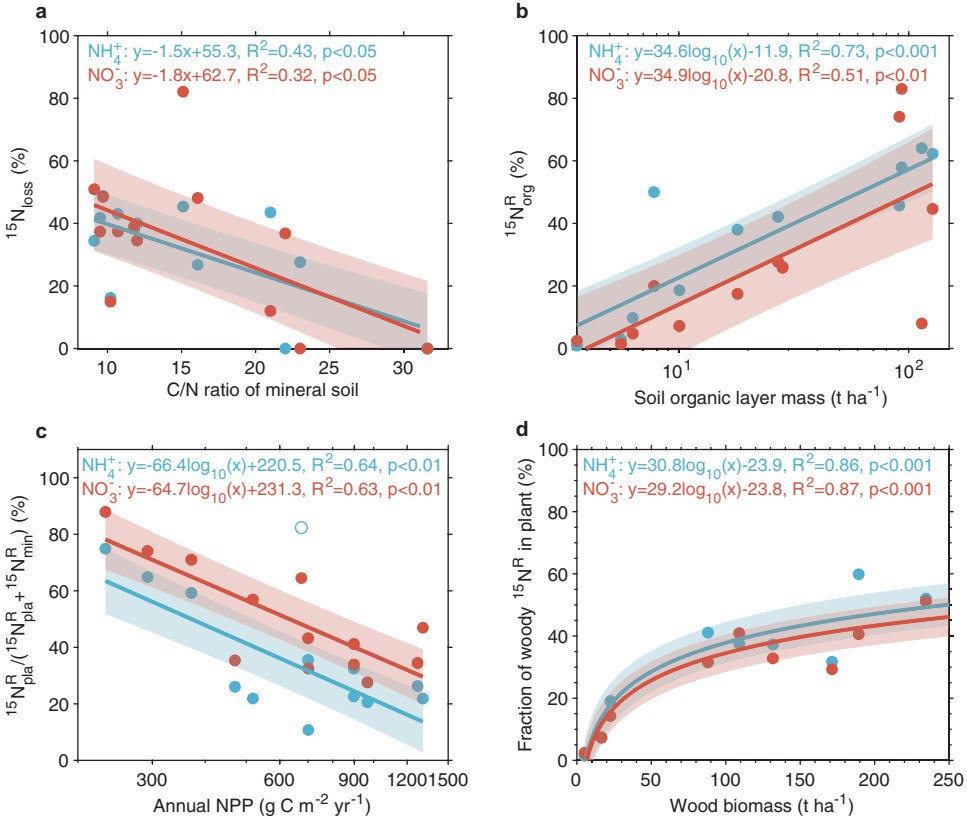

**Fig. 2 Relationships used in the scaling up of percent $^{15}$N-tracer recoveries in ecosystem pools or N leaching and gaseous losses. a** Percentages of $^{15}$N in leaching and gaseous losses as a function of C/N ratios of mineral soil (0–30 cm); **b** Percentages of $^{15}$N recovered in soil organic layer as a function of soil organic layer mass; **c** The relationship between $^{15}N_{pla}^{R}$ to the sum of $^{15}N_{pla}^{R}$ and $^{15}N_{min}^{R}$ and annual NPP; and **d** Percentages of $^{15}$N retained in plant that is allocated to woody biomass as a function of wood biomass. Data shown in each pane are mean values at each site. The corresponding shaded areas indicate the 95% confidence intervals of linear regressions. $^{15}N_{loss}$ indicates percentage of $^{15}$N leaching and gaseous losses which is the residual of 100 minus $^{15}$N recovery of ecosystem. $^{15}N_{pla}^{R}$ and $^{15}N_{min}^{R}$ indicate $^{15}$N recovery of plant and mineral soil pools, respectively, and the corresponding shaded areas indicate the 95% confidence intervals of linear regressions.

$2.0_{1.3}^{2.7}$ Tg N yr$^{-1}$ and $2.9_{2.0}^{3.7}$ Tg N yr$^{-1}$ of deposited NH$_x$ and NO$_y$, respectively, were retained in plant biomass (Supplementary Table 6). In woody biomass, $0.7_{0.5}^{1.0}$ Tg N yr$^{-1}$ of deposited NH$_x$ and $1.0_{0.7}^{1.2}$ Tg N yr$^{-1}$ of deposited NO$_y$ was retained. Retention in soil organic layer was $2.8_{2.3}^{3.6}$ Tg N yr$^{-1}$ for NH$_x$ and $2.4_{1.8}^{3.1}$ Tg N yr$^{-1}$ for NO$_y$ whereas retention in mineral soil was $4.6_{3.3}^{6.0}$ Tg N yr$^{-1}$ for NH$_x$ and $3.2_{2.4}^{4.3}$ Tg N yr$^{-1}$ for NO$_y$ (Supplementary Table 6).

The resulting spatial pattern of N retention indicates that N-limited forests in higher latitudes retain deposited N mostly in the vegetation and the organic soil, whereas in N-rich forests in lower latitudes the mineral soil is the dominant sink for deposited N (Fig. 3, Supplementary Fig. 7). The spatial analysis of N allocation clearly indicates that a similar fraction of deposited NH$_x$ is allocated to plants across biomes, while the retention fraction of deposited NO$_y$ in plants is larger in boreal forests than in temperate and tropical forests (Fig. 3). Biome-scale total retention per unit area of NH$_x$ and NO$_y$ is lower in tropical than in temperate and boreal forests (Fig. 3, Supplementary Table 6). Lower ecosystem N retention in tropical forests is likely due to higher gaseous losses and more leaching of deposited N (ref. [35]).

**Global carbon sink in forests from deposited NH$_x$ and NO$_y$.** Finally, we used our $^{15}$N-tracer-derived global maps of N retention in woody biomass and soils (Fig. 3, Supplementary Fig. 7) to estimate the C sink (annual net C gain) of global forests due to N deposition via the stoichiometric scaling method (see the Method). We estimated the total C sink due to N deposition in global forests

at $0.72_{0.49}^{0.96}$ Pg C yr$^{-1}$ (Fig. 4, Supplementary Table 7), which is 1.5–4 times the estimates from previous studies for forest ecosystems[19,36–41] and larger than most estimates for all terrestrial ecosystems (Supplementary Table 9). This larger estimate is partly because of the updated (and higher) N deposition values (23–27 vs 5–21 Tg N yr$^{-1}$) and partly because of greater C gain per unit N deposited obtained in this study ($29_{20}^{38}$ kg C kg$^{-1}$ N) than in the previous studies (Supplementary Table 9). Only the studies by Nadelhoffer et al.[19] and Thomas et al.[40] reported a higher C gain per unit N deposited, although their estimates of global N deposition (and the corresponding C sink) were low. Note that estimates by other approaches such as model analysis[39] account for the C sink over multi-decadal timescales. Our estimate based on retention of deposited N over 1 year reflects short-term (annual to decadal) effects of N deposition on C uptakes as the loss of previously deposited and retained N from the system could not be measured with the labelling technique. Our estimate indicates a likely ceiling for the N deposition contribution to the annual global terrestrial C sink over the last decade ($\approx$3.4 Pg C yr$^{-1}$)[42] of $21_{14}^{28}$% if past N deposition has had no effect on the C sink over time.

The global distribution of N-induced forest C sinks largely follows N deposition patterns, showing strong sinks in eastern North America, central Europe, South and East Asia, and parts of central Africa (Supplementary Fig. 9a). At biome scales, the largest N-induced C sink per unit area occurs in temperate forests (Supplementary Table 7). Although less NO$_y$ than NH$_x$ is deposited in global forests, it makes larger contributions to N-induced forest

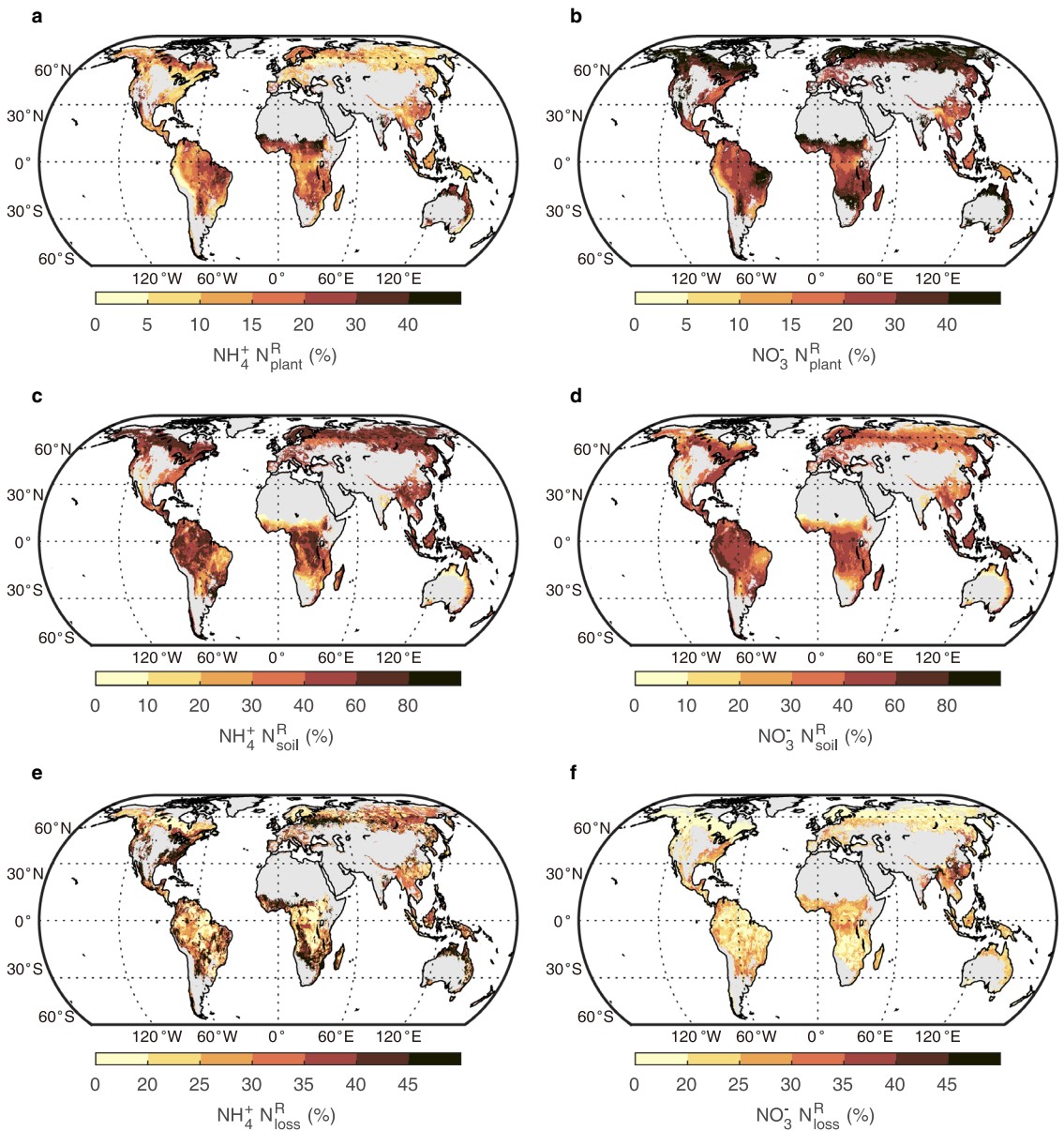

**Fig. 3 Spatial patterns of estimated N retention (%) in forest plants ($N^R_{plant}$) and soil ($N^R_{soil}$).** Figures **a**, **c** show the distribution of N retention for ammonium deposition while **b**, **d** show the distribution of N retention for nitrate deposition. Figure **e**, **f** show distribution of loss fraction ($^{15}N_{loss}$) for ammonium and nitrate depositions, respectively, and the loss fractions represents unrecovered $^{15}N$. Nitrogen retention by plants includes both trees and understory vegetation (shrubs, herbs, and grasses) while N retention in soil includes both soil organic layer and mineral soil. The N retention in plants is further divided into retention in woody (branches, stems, and coarse roots) versus non-woody tissues as shown in Supplementary Fig. 7.

C sink estimates at both the biome scale and the global scale because of higher C gain per unit N deposited for $NO_y$ ($35^{46}_{23}$ kg C kg$^{-1}$ N) than for $NH_x$ ($24^{33}_{17}$ kg C kg$^{-1}$ N) (Supplementary Table 7). This emphasizes the need to account separately for differential fates of reduced and oxidized N in deposition so that the contribution of deposited N to forest C sink and its implication for land C balance can be fully assessed.

Our global estimates have several limitations. First, the few paired $^{15}N$-labelling experiments are unevenly distributed. The absence of sites in African and South American forests reduces confidence in those regions' estimates. Our study sites were also located in regions with moderate and high ambient N deposition. Second, the $^{15}N$-labelling at most of our study sites was undertaken during the growing season when plants are likely to be highly active. The upscaling method assumes that deposited N

across the year is retained with the same efficiency, which likely overestimates plant N uptake and consequently the C sink. Third, the estimated C sink is based on the first-year fate of deposited N. Over a longer timescale, N loss or distribution and the turnover of C and N in plant tissues and soil organic matter may modify the effect of deposited N on C sink. Thus, our estimate likely provides an upper limit to the global forest C sink induced by N deposition.

In summary, by quantifying the differential fates of $^{15}NH_4^+$ and $^{15}NO_3^-$ tracers, we showed that N from deposited $NO_y$ is preferentially assimilated by plants, while N from deposited $NH_x$ is preferentially retained in soil. Deposited $NO_y$ is more effective than $NH_x$ at increasing C sinks in forest biomass due to its higher uptake by trees. We estimated the maximum potential contribution of deposited N to forest C sinks at $0.72^{0.96}_{0.49}$ Pg C yr$^{-1}$. Our

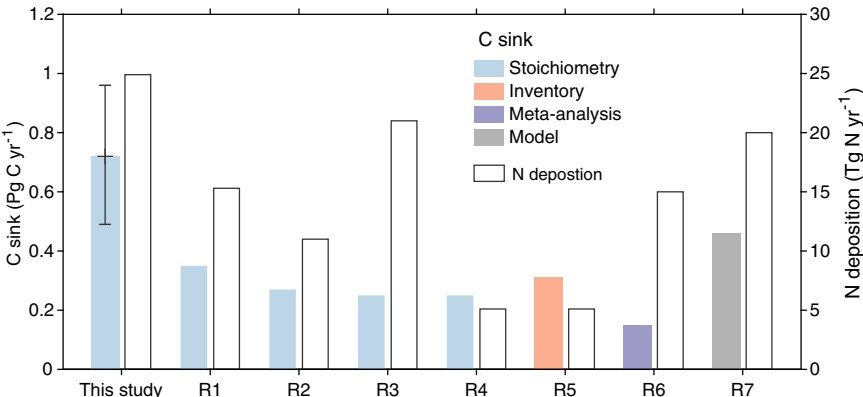

**Fig. 4 Estimates of C sink in global forests due to atmospheric N deposition.** In the current study, the C sink estimated due to N deposition (24.9 Tg N yr$^{-1}$) on lands with >10% forest cover assumes that 80% of atmospheric N retained in the soil is immobilized in the soil organic matter (see Method). The contribution of NH$_x$ and NO$_y$ deposition to the total N-induced C sink in the global forests is shown in Supplementary Table 7. The error bar for the current study indicates mean ranges with 95% confidence interval ($n$ = 24,000, derived from four N deposition data × 6 C/N data × 1000 Monte Carlo) (Method). Estimated C sinks induced by N deposition in global forest ecosystems reported by previous studies (R1-R7) which distinguished the effects of N deposition on C sinks are presented for comparison. References R1, R2, R3, R4, R5, R6, and R7 are De Vries et al.[36], Du and de Vries[37], Wang et al.[41], Nadelhoffer et al.[19], Thomas et al.[40], Schulte-Uebbing and De Vries[38], and Fleischer et al.[39], respectively.

findings imply future changes in the ratio of NH$_x$ and NO$_y$ inputs due to land use, technology, and policy changes will alter the retention pattern of deposited N and its potential contribution to C sinks in forest ecosystems.

## Methods

**Study sites.** The paired $^{15}$N-tracer experiments were conducted in 13 forest sites, of which nine were in China, two in Europe and two in the USA. These sites vary in mean annual precipitation (MAP) from 700 to 2500 mm, in mean annual temperature (MAT) from 3 to > 20 °C, and in soil types (Fig. 1, Supplementary Table 1, Supplementary Table 2). Ambient N deposition (bulk/throughfall NH$_4^+$ plus NO$_3^-$) at the sites ranged from 6 to 54 kg N ha$^{-1}$ yr$^{-1}$. Forest types at the experimental sites include tropical forests in southern China, subtropical forests in central China, and temperate forests in northeastern China, Europe, and the USA. Data from the sites in Europe, the USA, and six of the nine sites in China have been reported previously. Detailed descriptions of these sites and the related data source references are summarized in Supplementary Table 1. Data for forests at the other three sites in China (Xishuangbanna, Wuyishan, and Maoershan) are originally presented here. The Xishuangbanna sites, which is located Xishuangbanna National Forest Reserve in Menglun, Mengla County, Yunnan Province, is a primary mixed forest dominated by the typical tropical forest tree species *Terminalia myriocarpa* and *Pometia tomentosa*. The Wuyishan forest, which is located in the Wuyi mountains in Jiangxi Province, is also a mature subtropical forest with *Tsuga chinensis var. tchekiangensis* as the dominant tree species in the canopy layer. Other common tree species in the forest include *Betula luminifera* and *Cyclobalanopsis multinervis*. Maoershan is a relatively young (45 years) larch (*Larix gmelinii*) plantation located at Laoshan Forest Research Station of Northeast Forestry University, Heilongjiang Province. A few tree species *Juglans mandshurica*, *Quercus mongolica*, and *Betula platyphylla-* coexist with *Larix gmelinii* in the canopy. More information about these sites is also presented in Supplementary Table 1.

**$^{15}$N-tracer experiment.** At all sites, small amounts of $^{15}$NH$_4^+$ or $^{15}$NO$_3^-$ tracers (generally < 1% of the throughfall deposition) were added systematically to forest floors. In all the Chinese sites except Tieshanping, three replicate plots (20 m × 20 m) were each divided into two halves (10 m × 20 m), with one half receiving $^{15}$NH$_4^+$ and the other receiving $^{15}$NO$_3^-$. At Tieshanping, $^{15}$NH$_4^+$ or $^{15}$NO$_3^-$ were separately added to three replicate plots (14 m × 14 m). At Wülfersreuth, $^{15}$NH$_4^+$ or $^{15}$NO$_3^-$ were added to five replicate plots (40–70 m$^2$ plots) established in a young Norway spruce forest[43]. At Solling, three large replicate plots (300 m$^2$) established in a 72-year-old Norway spruce forest were divided into half, and one half was labelled with $^{15}$NH$_4^+$ and the other half was labelled with $^{15}$NO$_3^-$. In the $^{15}$N-tracer experiments at Harvard Forest in the US (a red pine forest and an oak-dominated deciduous forest), a single large plot (30 m × 30 m) in each forest was split into two, with one half of the plot receiving $^{15}$NH$_4^+$ and the other $^{15}$NO$_3^-$. $^{15}$N tracers were added once in all Chinese sites and at Wülfersreuth, and during two growing seasons (1991 and 1992) at Harvard Forest, and over 3 years (2002–2004) at Solling. Details of the $^{15}$N-tracer addition at each site are summarized in Supplementary Table 3.

**Sampling.** Major ecosystem compartments including trees, understory vegetation, fine roots, and organic and mineral soil layers were sampled before and ~1 year after

the $^{15}$N-labelling in each plot. Understory plants and tree compartments including mature leaves, twigs, branches, and stem woods were collected. Organic and mineral soil layers were sampled separately. For mineral soil, sampling depth varied among sites depending on the local soil conditions; 0–40 cm at all Chinese sites except at the Changbai forest (0–15 cm) and Tieshanping (0–30 cm), 0–65 cm at Wülfersreuth, 0–100 cm at Solling, and 0–20 cm at the two Harvard Forest sites.

**Chemical analysis.** At each site, analyses for $^{15}$N were conducted using isotope-ratio mass spectrometry on the two sets of plant and soil samples taken before and ~1 year after the $^{15}$N-labelling. The preparation of plant and organic layer samples consisted of oven-drying at 50–70 °C, while soil samples were mostly air-dried, but sometimes oven-dried at temperatures varying between 40–70 °C and then sieved (< 2 mm), varying with the site considered. The $^{15}$N content of all samples at all sites were analyzed using elemental analyser-isotope ratio mass spectrometry while using slightly different systems. Details on the variation in temperatures used in the preparation of the plant, organic layer, and soil samples and in measuring systems used are given in the supplementary material and related data source references.

**Calculation of $^{15}$N-tracer recoveries.** Percent recoveries ($^{15}$N$_{rec}$) of the added $^{15}$N tracers in each ecosystem compartment were estimated based on N pool size estimates and changes in $^{15}$N contents of ecosystem pools according to the principle of $^{15}$N mass balance[44] as shown by Eq. (1) below:

$$15_{N_{added}} + N_{pool-before} \times \%^{15}N_{atom-before} = 15_{N_{lost}} + N_{pool-after} \times \%^{15}N_{atom-after} \quad (1)$$

Where N$_{added}$ is the mass of $^{15}$N we experimentally added (kg N ha$^{-1}$); N$_{pool-before}$ and N$_{pool-after}$ are the N pool (kg N ha$^{-1}$) in each ecosystem compartment before and 1 year after $^{15}$N labelling, respectively; %$^{15}$N$_{atom-before}$ and %$^{15}$N$_{atom-after}$ are $^{15}$N abundance (%) before and after $^{15}$N labelling, respectively; $^{15}$N$_{lost}$ is the mass of the $^{15}$N experimentally added that is lost from the ecosystem.

From Eq. (1), we can derive Eq. (2) to calculate the mass of $^{15}$N recovered ($^{15}$N$_{rec}$) at ecosystem level as:

$$15_{N_{rec}} = \frac{N_{pool-after} \times \%^{15}N_{atom-after} - N_{pool-before} \times \%^{15}N_{atom-before}}{15_{N_{added}}} \quad (2)$$

Assuming that N pool did not change significantly over the study period, we can get Eq. (3) to calculate the $^{15}$N$_{rec}$ as per cent of the added $^{15}$N as follows:

$$15_{N_{rec}} = \frac{N_{pool-before} \times (\%^{15}N_{atom-after} - \%^{15}N_{atom-before})}{15_{N_{added}}} \times 100 \quad (3)$$

We used equation (3) to calculate the $^{15}$N-tracer recovery as we also did not account for net N increment in both plant and soil compartments change in N pool over the relatively short experimental period (about 1 year). Such a small net change in N pool is difficult to detect using the traditional inventory method, which requires repeated measurement on N pool during a longer time (usually every 5 years), especially for soil N pool due to its large background N pool and spatial heterogeneity.

**Upscaling of $^{15}$N recovery to global forest N retention.** We established conceptual pathways of $^{15}$N retention and partitioning after the $^{15}$N-labelling (Supplementary Fig. 1) and assumed that the unrecovered $^{15}$N is lost by leaching and

denitrification. To predict global N retention from results of [15]N recoveries in our paired [15]N-labelling experiments, we considered nine potential predictors from factors that influence ecosystem N retention as suggested in the literature[9,15,20,30,45] including variables that define climate (MAT, MAP), ecosystem N status or soil fertility (soil C/N ratio, soil clay content, leaf C/N ratio, N deposition), N pool (soil organic mass, wood biomass), and annual net primary production (NPP). Then, we fitted all possible regressions of [15]N retention in plant and soil pools and the [15]N loss fraction with the set of predictor variables across the 13 sites (Supplementary Table 4) using the glmulti package in R. Then, the best regression model was selected based on the minimum corrected Akaike information criterion[46], constrained by the cutoff of variance inflation factor (VIF) > 3 to avoid multicollinearity among predictors. Global maps of partitioning of deposited $NH_x$ and $NO_y$ (Fig. 1a) to plants, woody biomass and soil as well as loss patterns (Fig. 3, Supplementary Fig. 7) were derived based on the best regression models summarized in Supplementary Table 4, using globally gridded products of the corresponding predictors. In addition, the key variables with significant and the highest explanations for predicting variations in [15]N allocation to plant and soil pools and [15]N loss fraction across the 13 sites are shown in Fig. 2.

**Data sources for scaling up N retention**. A global map of mean annual NPP was obtained from MODIS NPP product (MOD17, version 5.5) for the period from 2000 to 2015, with a spatial resolution of 1 km[47]. We used C/N ratios of mineral soil at 0–10 cm from three global soil databases, (1) the Harmonized World Soil Database[48], (2) the gridded Global Soil Dataset for Earth System Modeling (GSDE) of Beijing Normal University (BNU)[49], and (3) the Global Observation-based Land-ecosystems Utilization Model of Carbon, Nitrogen, and Phosphorus (GOLUM-CNP v1.0) database[50]. Litter mass and C/N ratios of wood, organic, and mineral soil layers were obtained from GOLUM-CNP v1.0. Soil clay content data was also obtained from BNU.

**Global N deposition map**. The global map of average $NH_x$ and $NO_y$ deposition to forests between the years 2000 and 2010 was created using forest-specific values of $NH_x$ and $NO_y$ inputs obtained from four different models (Supplementary Table 5). This map of global N deposition was combined with a global forest cover map with a spatial resolution of 1 km that is derived from a consensus land-cover product[34]. Here, a forest cover fraction > 20% in a 1-km pixel was defined as forest. Based on this, we estimated the total global forest area to be ≈42 million km².

**Calculation of N-induced C sink**. The N-induced C sink was estimated via the stoichiometric upscaling method[19], i.e., by multiplying the N retention in woody tissues of stems, branches, and coarse roots and in the soil with the C/N ratios in these compartments. The C sink due to $NH_x$ and or $NO_y$ deposition was calculated separately using Eq. (4) as follows:

$$C_{sink} = N_{dep} \times \left( {}^{15}N_{org}^{R} \times \frac{C}{N_{org}} + {}^{15}N_{min}^{R} \times \frac{C}{N_{min}} + {}^{15}N_{wood}^{R} \times \frac{C}{N_{wood}} \times f \right) \quad (4)$$

where $N_{dep}$ is $NH_x$ or $NO_y$ deposition (kg N ha⁻¹ yr⁻¹); ${}^{15}N_{org}^{R}$, ${}^{15}N_{min}^{R}$ and ${}^{15}N_{wood}^{R}$ indicate the fraction of deposited $NH_x$ or $NO_y$ allocated to organic layer, mineral soil, and woody biomass, respectively; and $\frac{C}{N_{org}}$, $\frac{C}{N_{min}}$, and $\frac{C}{N_{wood}}$ indicate C/N ratios in the soil organic layer, soil mineral layer and woody plant biomass, respectively. $f$ is the fraction we applied to account for flexible C/N in response to elevated N deposition. At elevated N deposition, wood C/N ratio may decrease, and N accumulates without stimulating additional ecosystem C storage. To account for this scenario, we adopted a flexible stoichiometry[51], in which the effects of N deposition on wood C/N ratios are accounted for by multiplying the C/N ratios of wood with a fraction $f$ (from 1 to 0) depending on plant growth response to different rates of N deposition level (kg N ha⁻¹ yr⁻¹). Results of growth responses to experimental N addition and field N gradient studies show plant growth increased with increasing N deposition, flattening near 15–30 kg N ha⁻¹ yr⁻¹ and a reversal toward no enhanced growth response at about 100 kg N ha⁻¹ yr⁻¹ (ref. [36,52]). Therefore, for N deposition < 15 kg N ha⁻¹ yr⁻¹, we assumed that N deposition has no effect on C/N ratios of wood ($f$ = 1). Then, we assumed that $f$ decreases to 0.5 when N deposition reaches 30 kg N ha⁻¹ yr⁻¹ and assumed that $f$ decreases to 0 when N deposition reached 100 kg N ha⁻¹ yr⁻¹ (i.e., N deposition will not increase tree growth anymore, and no new C is gained due to N deposition).

In the soil, deposited N can be retained through assimilation into microbial biomass, immobilization in soil organic matter (SOM), and abiotic immobilization in the soil. It is the fraction of soil retained N that are immobilized in the persistent pool of SOM (as organic N) that can contribute to long-term C sinks in the soil. We assume the fraction to be ≈ 80% based on results from previous [15]N-tracer studies[53–56].

We used four sets of global N deposition (Supplementary Table 5), six different wood C/N values (Supplementary Table 8), and three different soil C/N values that varied with plant functional type[50] in our stochastic calculations of the N-induced C sink.

**Analysis of uncertainties in global N retention and associated C sink**. To estimate uncertainty of the global N retention map, we considered the uncertainty of regression coefficients and the input data for upscaling in Supplementary Fig. 8 for each grid. We randomly sampled 10 out of the 13 sites to do the regression and then upscaled with one randomly selected input dataset for 1000 times using Monte Carlo

methods to generate 1000 sets of global maps for retention or loss of deposited $NH_x$ and $NO_y$ (Fig. 3). The standard deviations of these 1000 sets of global maps are defined as the uncertainty of retention maps shown in Supplementary Fig. 8. For the uncertainties of the N-induced C sink, we considered the uncertainty of global retention maps (1000 sets), the uncertainty of global N deposition (four global N deposition maps; Supplementary Table 5), and the uncertainty of wood C/N ratios (six sets; Supplementary Table 8). Thus, we derived 24000 maps of N-induced C sink (4 global N deposition maps × 1000 sets of retention maps × 6 wood C/N ratios × 1 soil organic C/N ratio) for both deposited $NH_x$ and $NO_y$. The total uncertainty of the N-induced C sink is defined by 95% confidence intervals.

**Reporting summary**. Further information on research design is available in the Nature Research Reporting Summary linked to this article.

## Data availability

The raw N retention and C sink data used in this study are available in the Dryad Digital Repository (DOI: 10.5061/dryad.cfxpnvx3d) (ref. [57]). The MODIS NPP product (MOD17, version 5.5) is downloaded from https://lpdaac.usgs.gov/products/mod17a2hv061/. The C/N ratios of mineral soil at 0–10 cm were obtained from Harmonized World Soil Database (https://www.isric.org/documents/document-type/isric-report-201201-isric-wise-derived-soil-properties-5-5-arc-minutes), BNU soil dataset (http://globalchange.bnu.edu.cn/research/soilw#download), and the GOLUM-CNP (v1.0) (Wang et al.[50]). The global maps of modelled N deposition are obtained CICERO-OsloCTM2 (https://igacproject.org/activities/atmospheric-chemistry-climate-model-intercomparison-project-accmip), GISS-E2-1-G (https://igacproject.org/activities/atmospheric-chemistry-climate-model-intercomparison-project-accmip), THQ (https://esg.pik-potsdam.de/projects/isimip/), and EMEP (https://thredds.met.no/thredds/catalog/data/EMEP/Articles_data/Schwede_etal_Ndep_2018/catalog.html). The global forest cover map is obtained from http://www.earthenv.org/landcover.

## Code availability

The codes used in this work can be accessed by contacting the corresponding authors.

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

## Acknowledgements
The work was supported financially by the National Key Research and Development Program of China (2016YFA0600802, 2017YFC0212700), the Key Research Program of Frontier Sciences of Chinese Academy of Sciences (QYZDB-SSWDQC002), the National Natural Science Foundation of China (41773094, 41722101, U1602234), National Research Program for Key Issues in Air Pollution Control (DQGG0105-02) and K. C. Wong Education Foundation (GJTD-2018-07). We thank Dave Simpson (EMEP MSC-W, Norwegian Meteorological Institute, Oslo, Norway) for the forest-specific global N deposition data derived from the EMEP rv4.17 model.

## Author contributions
Y.F., P.G., G.A.G., S.P., W.D.V., W. Zhu, X.H. conceived and designed the study; A.W., S.L., E.B., T.S., D.C., W. Zho, Y.Z., Y.G., L.D. performed field experiment and laboratory analysis; G.A.G., S.P., W.D.V., A.W., S.L. performed data analysis; S.P., Y.X. conducted modelling; G.A.G., Y.F., W.D.V., S.P., O.L.P., P.G., P.C., E.A.H., W. Zhu, K.N., A.W., S.L. wrote the paper with contribution from J.Z., D.L., K.K., E.D., G.Z., S.H., and other co-authors.

## Competing interests
The authors declare no competing interests.
