## [Peer Review File · Nature Communications]

Title: Retention of deposited ammonium and nitrate and its impact on the global forest carbon sinksREVIEWER COMMENTS

Reviewer #1 (Remarks to the Author):

Prior reviews of earlier versions of this manuscript have nicely summarized its insights and importance, along with a number of concerns. The authors have made substantial revisions to their earlier manuscript in response to these reviewer concerns, this time with considerable reduction of the estimated global C sinks extrapolated from ^{15}N tracer studies at 13 sites. I'm glad to see that section reduced in emphasis and presented with more caution, along with other revisions discussing the direct ^{15}N results.

I am happy to support nearly all of the current draft for publication in Nature Communications.

A few smaller concerns remain:

Line 60-61. The conclusion of the Abstract gives the impression that the "higher than previous estimates" of N deposition-induced global C sink in forests is mostly due to the insights from the reported ^{15}N field experiments, when careful reading of the main text & Supplement indicate a dominant role of the (appropriately) much larger global rate of N deposition than considered previously. Include that important driver somewhere when summarizing that global C sink in the abstract: "We estimate the N deposition-induced C sink in forests at 0.72 (0.49 – 0.96) Pg C yr⁻¹, higher than previous estimates" Indicate why: e.g.,because of both a larger role for oxidized N and 1.5-4 times greater rate of global N deposition than considered previously. (or similar)

Line 141-142. Greater anion adsorption capacity of tropical soils cannot readily explain greater retention of both NH_4^+ (cation) and NO_3^- (anion) in mineral soils compared to temperate sites, as written here. Rephrase, or reconsider this explanation. An alternative seemingly implied in the next line (144): tropical soils retain more ^{15}N in mineral soil because their smaller organic layers retain less ^{15}N than in temperate sites.

Line 165-168. The reported correlation between NPP and the ratio of ^{15}N retention in plants / (plants + mineral soil) still lacks a clear mechanistic interpretation, and it's not apparent whether it is a direct meaningful correlation, or an result of statistical covariation among other variables (organic and mineral soil ^{15}N retention; NPP; temperature). Line 169: "may also imply" (instead of "implies")

Line 218. This statement, that N deposition "likely accounts for 21% (14-28%) of the global terrestrial C sink over the last decade" appears overly confident on this specific fractional contribution given the many caveats and limitations that now receive greater recognition elsewhere in the text and Authors' response. Encourage "could be" (or similar) vs "likely".

Line 300. Accumulation of (plant) biomass is not the same thing as net primary production (plant biomass accumulation + biomass production & turnover), as indicated here. Clarify which term is meant.

Line 346-7. Rephrase: If N remains in the inorganic extractable N pool, it has not been “immobilized”.

Reviewer #3 (Remarks to the Author):

I reviewed the previous versions of this manuscript in Nature and was impressed about the quality and novelty since the beginning. Version after version, the manuscript has been improving, specially in their treatment of uncertainties and making the paper more balanced, putting more emphasis in the most robust parts of the analysis instead of the flashy but uncertain results. I was surprised the paper wasn't accepted in Nature despite all the improvements. In this new version, the authors have carefully considered all my previous comments, and I consider the paper ready for acceptance. This is a very important and exciting paper and I strongly recommend publication. Below are some minor comments:

- After so many important changes, the abstract has lost some of its original shine. I suggest better highlighting the rather striking result that nitrate is retained in vegetation more strongly than ammonium, contrary to the historical dogma.
- Indistinct use of NH_4/NO_3 , ammonium/nitrate, reduced/oxidized forms. I suggest sticking with one format throughout the manuscript.
- Typos: L169, L232, I suggest proofreading.
- L217: “likely” is an overstatement, we can't know if the result is accurate, and it actually isn't. The real value could be above or below the range.

Best wishes,
César Terrer, MIT (signed review)

Reviewer #4 (Remarks to the Author):

Gurmesa and Wang et al have made a good effort to incorporate the various second-round reviewer comments, especially using multiple variables in the regressions and improving the quantification of uncertainty. I'm still not in agreement that the method support the conclusion. This is perhaps simply an issue of semantics.

Although the authors managed to mostly convince me in the previous rounds of review that their under-sampling of mortality is OK for the timescale of inference – annual to at best a decade – the authors have not done a good enough job of communicating this timescale.

At best I think what the authors have done is provide an upper limit on the longer-term global sink. Long-term retention of the ^{15}N and associated carbon is unknown using this method.

As I said in my previous review “over longer timescales there’s turnover of vegetative tissue and soil organic matter and it’s likely over time the retained N over a year is likely to lose some additional unquantified fraction. Model estimates of the N deposition effect on the C sink will account for this. [...] The method used accounts for the annual net effect of N deposition on the C sink, but you cannot say anything about the net effect of N deposition cumulated over multiple decades.”

The authors cite two papers (line 240) to infer that longer-term the retention is not changed. However, these studies are based on 3 years and 6 years of data. This is not long enough to account for the turnover times of organic matter in ecosystems and quantify “sequestration” or long term sinks. Please quantify the timescale of these studies cited on line 240.

The word sequestration is used throughout, when really what is meant is annual net carbon gain. Sequestration implies a long timescale, how long is debatable but is certainly longer than a year, or a decade. Please change this language to better reflect the timescale of inference.

I find the title misleading:

“Retention of deposited ammonium and nitrate and its impact on the global forest carbon sinks.”

I think a more accurate title would be something like:

Short-term retention of deposited ammonium and nitrate provides an upper limit on the deposition-induced global forest carbon sink.

And the conclusion on line 245-246 would be more accurately stated: “We estimated the maximum potential contribution of deposited N to forest C sinks at $0.72 \text{ Pg C yr}^{-1}$, greater than previously acknowledged.”

The comparison with previous/other estimates (Fig , lns 213-218) requires more interpretation of the differences in the context of timescales. As I’ve said, models will account for the effects of long-term deposition and ecosystem turnover time that the present study cannot. This needs to be stated in the discussion of these results around ln 218.

These kinds of semantic arguments would normally lead me to recommend minor revisions but I think they’re so fundamental to preventing mis-interpretation and we need to do a better job of nuance and accuracy in our inferences in this important field. So I’m recommending major revisions.

Replies to the referees on “Retention of deposited ammonium and nitrate and its impact on global forest carbon sinks” by Geshere et al. (NCOMMS-21-30831-T).

Reviewer #1 (Remarks to the Author):

Prior reviews of earlier versions of this manuscript have nicely summarized its insights and importance, along with a number of concerns. The authors have made substantial revisions to their earlier manuscript in response to these reviewer concerns, this time with considerable reduction of the estimated global C sinks extrapolated from ¹⁵N tracer studies at 13 sites. I’m glad to see that section reduced in emphasis and presented with more caution, along with other revisions discussing the direct ¹⁵N results.

I am happy to support nearly all of the current draft for publication in Nature Communications.

Response: We thank the reviewer for the additional comments, suggestions, and corrections, which have helped us improve the manuscript.

A few smaller concerns remain:

Line 60-61. The conclusion of the Abstract gives the impression that the “higher than previous estimates” of N deposition-induced global C sink in forests is mostly due to the insights from the reported ¹⁵N field experiments, when careful reading of the main text & Supplement indicate a dominant role of the (appropriately) much larger global rate of N deposition than considered previously. Include that important driver somewhere when summarizing that global C sink in the abstract: “We estimate the N deposition-induced C sink in forests at 0.72 (0.49 - 0.96) Pg C yr⁻¹, higher than previous estimates” Indicate why: e.g., ...because of both a larger role for oxidized N and 1.5-4 times greater rate of global N deposition than considered previously. (or similar)

Response: Accepted. We have indicated the reason why our estimate is higher than previous one (the larger role of oxidized N and increased N deposition) in the abstract, as suggested (line 61-62).

Line 141-142. Greater anion adsorption capacity of tropical soils cannot readily explain greater retention of *both* NH₄⁺ (cation) and NO₃⁻ (anion) in mineral soils compared to temperate sites, as written here. Rephrase, or reconsider this explanation. An alternative seemingly implied in the next line (144): tropical soils retain more ¹⁵N in mineral soil because their smaller organic layers retain less ¹⁵N than in temperate sites.

Response: We agree. We have revised the sentences about the retention of ammonium and nitrate in temperate vs. tropical mineral soil (lines 143-150). We have now indicated that the greater anion adsorption capacity of tropical soils explains great retention of nitrate in mineral soils compared to temperate sites as follow:

"In addition, a greater anion adsorption capacity of highly weathered tropical mineral soils could increase retention of the highly leachable nitrate in tropical mineral soil" (line 149-150).

Line 165-168. The reported correlation between NPP and the ratio of ¹⁵N retention in plants / (plants + mineral soil) still lacks a clear mechanistic interpretation, and it’s not apparent whether

it is a direct meaningful correlation, or a result of statistical covariation among other variables (organic and mineral soil 15N retention; NPP; temperature).

Response: We have edited the text to clarify that the correlation could also indicate indirect climatic control on fates of deposited N in forests (line 174-176).

Line 169: “may also imply” (instead of “implies”)

Response: Corrected.

Line 218. This statement, that N deposition “likely accounts for 21% (14-28%) of the global terrestrial C sink over the last decade” appears overly confident on this specific fractional contribution given the many caveats and limitations that now receive greater recognition elsewhere in the text and Authors’ response. Encourage “could be” (or similar) vs “likely”.

Response: We agree. We have revised the sentence as follow:

"In this regard, our estimate indicates that N deposition could account for 21₁₄²⁸% of the annual global terrestrial C sink ($\approx 3.4 \text{ Pg C yr}^{-1}$) over the last decade" (lines 231-232).

Line 300. Accumulation of (plant) biomass is not the same thing as net primary production (plant biomass accumulation + biomass production & turnover), as indicated here. Clarify which term is meant.

Response: We have clarified that we meant net primary production (NPP) (lines 314-315).

Line 346-7. Rephrase: If N remains in the inorganic extractable N pool, it has not been “immobilized”.

Response: We have deleted the phrase "(i.e., N remains in the extractable pool as inorganic N)" (lines 361).

Reviewer #3 (Remarks to the Author):

I reviewed the previous versions of this manuscript in Nature and was impressed about the quality and novelty since the beginning. Version after version, the manuscript has been improving, especially in their treatment of uncertainties and making the paper more balanced, putting more emphasis in the most robust parts of the analysis instead of the flashy but uncertain results. I was surprised the paper wasn't accepted in Nature despite all the improvements. In this new version, the authors have carefully considered all my previous comments, and I consider the paper ready for acceptance. This is a very important and exciting paper and I strongly recommend publication.

Response: We thank Dr. César Terrer for his assessment of our manuscript and giving us invaluable comments and suggestions.

Below are some minor comments:

- After so many important changes, the abstract has lost some of its original shine. I suggest better highlighting the rather striking result that nitrate is retained in vegetation more strongly than ammonium, contrary to the historical dogma.

Response: We have tried to refine the abstract following the collective suggestions from you and the other reviewers.

- Indistinct use of NH₄/NO₃, ammonium/nitrate, reduced/oxidized forms. I suggest sticking with one format throughout the manuscript.

Response: We have changed the chemical formula (NH₄/NO₃) to full names (ammonium/nitrate) throughout the manuscript. We have also consistently used 'ammonium and nitrate' to discuss ¹⁵N retention. However, we still used 'reduced vs oxidized' to discuss C sink because we calculated the C sink is based on all forms of reduced and oxidized N whose retention fraction was inferred from the fate of NH₄ and NO₃, respectively.

- Typos: L169, L232, I suggest proofreading.

Response: We have corrected the typos.

- L217: "likely" is an overstatement, we can't know if the result is accurate, and it actually isn't. The real value could be above or below the range.

Response: We have revised the statement as follow (using 'could' as suggested by R#1):

"In this regard, our estimate indicates that N deposition could account for $21\frac{28}{14}\%$ of the annual global terrestrial C sink ($\approx 3.4 \text{ Pg C yr}^{-1}$) over the last decade" (lines 231-232).

Best wishes,
César Terrer, MIT (signed review)

Reviewer #4 (Remarks to the Author):

Gurmesa and Wang et al have made a good effort to incorporate the various second-round reviewer comments, especially using multiple variables in the regressions and improving the quantification of uncertainty. I'm still not in agreement that the method support the conclusion. This is perhaps simply an issue of semantics.

Response: We thank the reviewer for taking time to evaluate our revised manuscript and give us detailed comments on the need to clearly communicate the timescale over which our analysis is applicable.

Although the authors managed to mostly convince me in the previous rounds of review that their under-sampling of mortality is OK for the timescale of inference – annual to at best a decade – the authors have not done a good enough job of communicating this timescale. At best I think what the authors have done is provide an upper limit on the longer-term global sink. Long-term retention of the 15N and associated carbon is unknown using this method.

As I said in my previous review “over longer timescales there’s turnover of vegetative tissue and soil organic matter and it’s likely over time the retained N over a year is likely to lose some additional unquantified fraction. Model estimates of the N deposition effect on the C sink will account for this. [...] The method used accounts for the annual net effect of N deposition on the C sink, but you cannot say anything about the net effect of N deposition cumulated over multiple decades.”

The authors cite two papers (line 240) to infer that longer-term the retention is not changed. However, these studies are based on 3 years and 6 years of data. This is not long enough to account for the turnover times of organic matter in ecosystems and quantify “sequestration” or long term sinks. Please quantify the timescale of these studies cited on line 240.

The word sequestration is used throughout, when really what is meant is annual net carbon gain. Sequestration implies a long timescale, how long is debatable but is certainly longer than a year, or a decade. Please change this language to better reflect the timescale of inference.

Response: We now agree with the above perspectives raised by the reviewer, and we have made several changes in the revised manuscript to address it. First, we have indicated the timescale over which our analysis is applicable as 'annual to decadal' (lines 226-231) and cited more long-term studies (decadal to multi-decadal) to support it. Second, we have revised the sentence “However, a few multi-year studies indicate that the redistribution of N is minor” to indicate that our estimate is likely an upper limit due to N loss and redistribution over a longer timescale as follow:

"Thus, our estimate likely provides an upper limit on the deposition-induced forest C sink" (line 254).

Third, we have replaced the term “sequestration” with “sink” in the revised manuscript and mentioned that it refers to annual net C gain (line 217). We used 'Sequestration' only when referring to 'C sequestration efficiency' of deposited N, but we have indicated it means 'kg C per kg deposited N' in our study (line 223).

I find the title misleading:

“Retention of deposited ammonium and nitrate and its impact on the global forest carbon sinks.”

I think a more accurate title would be something like:

Short-term retention of deposited ammonium and nitrate provides an upper limit on the deposition-induced global forest carbon sink.

Response: We wish to keep the original title, as it is short and covers our two main objectives: 1) to quantify the ^{15}N fate and distribution in forest ecosystem on a global scale; 2) to estimate C sequestration contributed by N deposition for global forests. The title suggested by the reviewer may have too much weight on the second objective, while we would like to balance the weight between the first and second objectives. Furthermore, Nature Communications requires a title without an active verb (e.g., 'provides'). Yet, we are certainly open to further suggestions from the reviewer and the editor.

And the conclusion on line 245-246 would be more accurately stated: “We estimated the maximum potential contribution of deposited N to forest C sinks at $0.72 \text{ Pg C yr}^{-1}$, greater than previously acknowledged.”

Response: Done (lines 249-250).

The comparison with previous/other estimates (Fig , lns 213-218) requires more interpretation of the differences in the context of timescales. As I've said, models will account for the effects of long-term deposition and ecosystem turnover time that the present study cannot. This needs to be stated in the discussion of these results around ln 218.

Response: Done as suggested. We have stated it as follow:

"Note that estimates by other approaches such as model analysis³⁷ account for C sink over multi-decadal time scale. Our estimate based on retention of deposited N over one year likely reflects annual to decadal effects of N deposition on C sinks given the retention (and accumulating in wood) of N from deposition in forests for up to sever decades once it stays in the plant and soil pools over the first growing season³⁸⁻⁴¹" (line 226-231).

Please note that we have now cited long-term studies (decadal) from field ^{15}N tracer experiment and model analysis of time series of fates of deposited N (Ref³⁸⁻⁴¹).

These kinds of semantic arguments would normally lead me to recommend minor revisions but I think they're so fundamental to preventing mis-interpretation and we need to do a better job of nuance and accuracy in our inferences in this important field. So I'm recommending major revisions.

Response: Thank you once again for your careful and professional evaluations.

REVIEWER COMMENTS

Reviewer #4 (Remarks to the Author):

Ln 50-51: Please change to “To determine one-year forest N retention fractions for deposited ammonium and nitrate,”

Ln 58 and throughout: “Carbon sequestration efficiency” is not correct. This carbon is cannot be considered sequestered after one year. If this terminology has been used in the past it is incorrect, please change to something more accurate.

Ln 60-62. Please add some text that describes the additional reason for the over-estimate is that the method does not account for all fluxes (i.e. carbon losses of gains from previous N deposition) and so is not truly the net sink.

Ln 228-230. “Our estimate based on retention of deposited N over one year likely reflects annual to decadal effects of N deposition on C sinks given the retention (and accumulating in wood) of N from deposition in forests for up to sever decades once it stays in the plant and soil pools over the first growing season.” This doesn’t fully make sense, nor acknowledge the unmeasured N-loss fluxes from the system of previously deposited N that was taken up. Please change to: “Our estimate based on retention of deposited N over one year reflects short-term (annual to decadal) effects of N deposition on C sinks given the loss of previously deposited and retained N from the system could not be measured with the labeling technique.”

Ln 231-232: Again, the authors are confusing a short-term, small-scale sink from the true net sink and the global scale. Please change to: “In this regard, our estimate indicates a ceiling for the N deposition contribution to the annual global terrestrial C sink over the last decade ($\approx 3.4 \text{ Pg C yr}^{-1}$) of 21%.” It could be added that this ceiling would only be realized if past N deposition has had no effect on the C sink.

Ln 260: delete “greater than previously acknowledged.” The estimate is different from some of the previous estimates while this line suggests an apples to apples comparison when that is not the case. As I said above, this statement only holds true if you assume that past N deposition has had no effect on the C sink. Without being able to quantify the losses of previously retained deposited N the true net sink cannot be measured/estimated.

Suggest deleting ln 260-261, contains little useful information.

Replies to the referees on “Retention of deposited ammonium and nitrate and its impact on global forest carbon sinks” by Gurmesa et al. (NCOMMS-21-30831A).

Reviewer #4 (Remarks to the Author):

Ln 50-51: Please change to “To determine one-year forest N retention fractions for deposited ammonium and nitrate,”

Response: Done as suggested.

Ln 58 and throughout: “Carbon sequestration efficiency” is not correct. This carbon is cannot be considered sequestered after one year. If this terminology has been used in the past it is incorrect, please change to something more accurate.

Response: Done as suggested.

Ln 60-62. Please add some text that describes the additional reason for the over-estimate is that the method does not account for all fluxes (i.e., carbon losses of gains from previous N deposition) and so is not truly the net sink.

Response: Done as suggested.

Ln 228-230. “Our estimate based on retention of deposited N over one year likely reflects annual to decadal effects of N deposition on C sinks given the retention (and accumulating in wood) of N from deposition in forests for up to sever decades once it stays in the plant and soil pools over the first growing season.” This doesn’t fully make sense, nor acknowledge the unmeasured N-loss fluxes from the system of previously deposited N that was taken up. Please change to: “Our estimate based on retention of deposited N over one year reflects short-term (annual to decadal) effects of N deposition on C sinks given the loss of previously deposited and retained N from the system could not be measured with the labeling technique.”

Response: Done as suggested.

Ln 231-232: Again, the authors are confusing a short-term, small-scale sink from the true net sink and the global scale. Please change to: “In this regard, our estimate indicates a ceiling for the N deposition contribution to the annual global terrestrial C sink over the last decade (≈ 3.4 Pg C yr⁻¹) of 21%.” It could be added that this ceiling would only be realized if past N deposition has had no effect on the C sink.

Response: Done as suggested.

Ln 260: delete “greater than previously acknowledged.” The estimate is different from some of the previous estimates while this line suggests an apples to apples comparison when that is not the case. As I said above, this statement only holds true if you assume that past N deposition has had no effect on the C sink. Without being able to quantify the losses of previously retained deposited N the true net sink cannot be measured/estimated.

Response: Done as suggested.

Suggest deleting ln 260-261, contains little useful information.

Response: We have replaced the sentence by the following to specify the implication of our study:

"Our findings imply future changes in the ratio of NH_x and NO_y inputs due to land use, technology, and policy changes will alter the retention pattern of deposited N and its potential contribution to C sink in forest ecosystems."